# Learning Semantics-Aware Locomotion Skills from Human Demonstration

**Yuxiang Yang**[12], **Xiangyun Meng**[1], **Wenhao Yu**[2], **Tingnan Zhang**[2], **Jie Tan**[2], **Byron Boots**[1]
[1]University of Washington, [2]Robotics at Google
{yuxiangy,xiangyun,bboots}@cs.washington.edu
{magicmelon,tingnan,jietan}@google.com

**Abstract:** The semantics of the environment, such as the terrain types and properties, reveal important information for legged robots to adjust their behaviors. In this work, we present a framework that uses semantic information from RGB images to adjust the speeds and gaits for quadrupedal robots, such that the robot can traverse through complex offroad terrains. Due to the lack of high-fidelity offroad simulation, our framework needs to be trained directly in the real world, which brings unique challenges in sample efficiency and safety. To ensure sample efficiency, we pre-train the perception model on an off-road driving dataset. To avoid the risks of real-world policy exploration, we leverage human demonstration to train a speed policy that selects a desired forward speed from camera images. For maximum stability, we pair the speed policy with a gait selector, which selects a robust locomotion gait for each forward speed. Using only 40 minutes of human demonstration data, our framework learns to adjust the speed and gait of the robot based on perceived terrain semantics and enables the robot to walk over 6km safely and efficiently.

**Keywords:** Legged Locomotion, Semantic Perception, Imitation Learning, Hierarchical Control

## 1 Introduction

To operate in complex offroad environments, it is crucial for quadrupedal robots to adapt their motion based on the perception of the terrain ahead. When encountering new terrains, the robot needs to identify changes in key terrain properties, such as friction and deformability, and respond with the appropriate locomotion strategy to maintain a reasonable forward speed without incurring failures. In many cases, information about such terrain properties is more easily inferred from a terrain's *semantic* class (e.g. grass, mud, asphalt, etc.) instead of its *geometric* shape (e.g. slope angle, smoothness) [1, 2]. However, recent works in perceptive locomotion [3, 4, 5, 6, 7, 8, 9] mostly focus on the *geometric* aspect of the terrain, and do not make use of such *semantic* information.

In this work, we present a framework for quadrupedal robots to adapt locomotion behaviors based on perceived terrain semantics. The central challenge in learning such a semantic-aware locomotion controller is the high cost of data collection. On the one hand, while simulation has become an effective data source for many robot learning tasks, modeling the complex contact dynamics accurately and rendering photorealistic offroad terrains is not yet possible in simulation. On the other hand, data collection in the real world is time-consuming and requires significant human labor. Moreover, the robot needs to remain safe during the data collection process, as any robot failure can cause significant damage to the hardware and surrounding environment. Therefore, it is difficult to use standard reinforcement learning methods for this task.

Our framework addresses all concerns above, and learns semantics-aware locomotion skills directly in the real world. To reduce the amount of data required, we pre-train a semantic segmentation network on an off-road driving dataset and extract a semantic embedding from the model for further fine-tuning. To avoid policy exploration in real-world environments, we collect speed choices from human demonstrations and train the policy using imitation learning [10]. Additionally, inspired by previous results on the relationship between speed and gait in animals [11] and legged robots

6th Conference on Robot Learning (CoRL 2022), Auckland, New Zealand.

[12, 13], we pair the speed policy with a gait selector to further improve the robot's stability. With the pre-trained image embedding, the imitation learning setup, and the gait selector, our framework learns semantics-aware locomotion skills directly in the real world safely and efficiently.

We deploy our framework on an A1 quadrupedal robot from Unitree [14]. Using only 40 minutes of human demonstration data, our framework learns semantics-aware locomotion skills that can be directly deployed for offroad missions. The learned skill policy inspects the environment and selects a fast and robust locomotion skill for each terrain, from slow and cautious stepping on heavy pebbles to fast and active running on flat asphalts. The learned framework generalizes well and operates without failure on a number of trails not seen during training (over 6km in total). Moreover, our framework outperforms the manufacturer's default controller in terms of speed and safety. We further conduct ablation studies to justify the important design choices.

The technical contributions of this paper include the following:

1. We develop a hierarchical framework that adapts locomotion skills from terrain semantics.

2. We propose a safe and data-efficient method to train our framework directly in the real world, which only requires 40 minutes of human demonstration data.

3. We evaluate the trained framework on multiple trails spanning 6km with different terrain types, where the robot reached high speed and walked without failures.

## 2   Related Works

**Perception for Legged Robots**   Creating a perceptive locomotion controller is a critical step to enable legged robots to walk in offroad, unstructured environments. Most importantly, it allows robots to detect and react to terrain changes proactively before contact. Many prior works have focused on understanding terrain *geometry* from perceptive sensors [15, 16, 3, 7, 6]. However, such information can be insufficient as it does not reveal important terrain properties such as deformability or contact friction [8, 9, 17, 18]. To ameliorate this, recent works proposed to update this geometric understanding of terrain with proprioceptive information [8, 9, 17]. However, these methods sacrifice proactivity, as the update cannot happen until *after* the robot has stepped on the terrain.

Another approach is to infer the terrain properties from its *semantics* [19, 20, 21, 22] so that the robot can detect changes in terrain property *before* contact, and select its locomotion strategies proactively. Recently, Suryamurthy et al. [23] trained models to predict terrain class and roughness and used the prediction to modulate the height and navigational path of a wheel-leg hybrid robot in an indoor environment. Our framework uses a similar semantics-based approach in the perception module and extends the result to off-road environments with a wide variety of terrains by adapting both the speed and gait of the robot.

**Terrain Traversability Estimation**   The goal of our perception module is to assess the traversability of the terrain ahead of the robot. Researchers have proposed a number of approaches to estimate traversability from perception data, including manually designed [24], learned from self-exploration [16, 25], or learned from human demonstration [26]. While learning-based approaches provide more flexibility, they usually require large amounts of data, which is difficult to collect in the real world. As a result, most approaches rely on simulation [27] as a source for training data. However, simulation is not feasible for our task, as it is currently difficult to accurately model the complex contact dynamics and create photorealistic renderings of off-road environments. Unlike previous approaches, our framework can be trained directly in the real world and requires only 40 minutes of human demonstration data.

**Motion Controller Design for Perceptive Locomotion**   Another important question in perceptive locomotion is the design of a motion controller that effectively makes use of the perceptive information. A common strategy is to create a low-level motion controller that plans precise foothold placements based on the perceived terrain [3, 4, 5, 6, 7]. While these methods have shown good results in highly uneven terrains, the high computational cost required for terrain understanding and rapid planning makes it infeasible for complex offroad environments. In this work, we devise a novel way to interface between perception and low-level motion controllers for legged robots, where the high-level perception model outputs the desired locomotion skills, including forward speed and

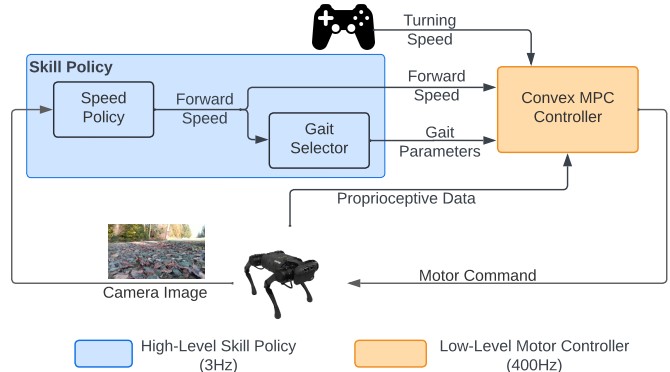

Figure 1: Our framework consists of a high-level skill policy and a low-level motor controller. The skill policy selects locomotion skills (gait and speed) based on camera images. The low-level controller computes motor commands for robot control.

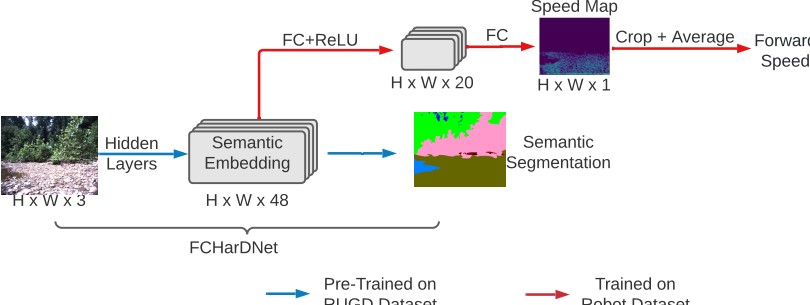

Figure 2: Architecture of our perception model. We extract a semantic embedding from a pre-trained semantic segmentation network and use it to learn and predict forward speeds.

robot gait, to a low-level motor controller. With our framework, the robot can select a safe and fast walking strategy for different terrains, which is crucial for offroad traversal.

# 3 Overview

Our hierarchical framework (Fig. 1) consists of a high-level skill policy and a low-level motor controller. At the high level, the skill policy receives the RGB image stream from the onboard camera and determines the corresponding locomotion skill. Each skill consists of a desired forward speed and a corresponding locomotion gait, which are computed by the speed policy and gait selector, respectively. We train the speed policy using imitation learning from human demonstrations and manually design the gait selector to find the appropriate gait for each forward speed. At the low level, a convex MPC controller [28] receives the skill command from the skill policy and computes motor commands for robot control. In addition, the convex MPC controller can optionally receive a steering command from an external teleoperator, which specifies the desired turning rate.

# 4 Learning Speed Policies

In unstructured offroad terrains, it is crucial for a robot to adjust its speed in response to terrain changes so that it can traverse through different terrains efficiently and without failure. To achieve that, we design a speed policy, which computes the desired forward speed of the robot based on camera images. We train the speed policy using a two-staged procedure: First, we pre-train a semantic embedding from an offline dataset. After that, we collect human demonstrations and train the speed policy using imitation learning.

## 4.1 Pre-trained Semantic Embedding

To reduce the amount of real-world data required to train the speed policy, we pre-train a semantic segmentation model and extract a semantic embedding for subsequent finetuning. We implement the model based on FCHarDNet-70 [29], which is a compact fully-convolutional encoder-decoder architecture with good real-time performance. We pre-train the model on the RUGD dataset [30], an off-road driving dataset with pixel-wise semantic labels (grass, dirt, rock, etc.). We choose RUGD because of its similarity to the images collected by the robot camera.

The next step is to extract an embedding from the pre-trained FCHarDNet [29] model for finetuning on robot data. Although the pre-trained model performs well on the RUGD dataset, its predicted segmentation becomes less accurate on robot images due to distribution shift. Meanwhile, the output of the hidden layers still provides a continuous semantic description for each pixel. Therefore, we extract a *semantic embedding* from the output of the last hidden layer in FCHarDNet, which assigns a 48-dimensional embedding vector to each pixel in the input image (Fig. 2). We then compute a speed map by feeding the embedding of each pixel through a fully-connected layer and compute the forward speed by averaging over a fixed region at the bottom of the speed map, which roughly corresponds to a rectangular area 1m long, 0.3m wide in front of the robot. The speed map provides a straightforward and intuitive way to understand the model's predictions and can be used in navigational tasks such as path planning.

## 4.2 Learning Speed Commands from Human Demonstration

Even with the pre-trained semantic embedding, finding the appropriate speeds for offroad terrains using reinforcement learning is still challenging due to omnipresent noise and safety concerns in the real world. As an alternative, a human operator can readily assess the robot's stability and adjusts the speed command accordingly based on the operator's previous experience with the robot platform. Therefore, we collect speed commands from human demonstrations and train the speed policy using imitation learning.

We collect human demonstrations by tele-operating the robot on a variety of terrains, including asphalt, pebble, grass and dirt. During data collection, the operator gives speed commands using a joystick, while other components of the pipeline, such as the gait selector and motor controller, function accordingly (Fig. 1). Each time the camera captures a new image, we store the image and the corresponding speed command. We then train the speed policy using behavior cloning [10], where the objective is to minimize the difference between predicted speed and human command.

## 5 Speed-based Gait Selector

In addition to speed, the *gait* of a legged robot, such as its foot swing height, can greatly affect its traversability, especially on uneven terrains. While the perception policy can output speed and gait parameters jointly, training such a policy using imitation learning can be challenging, as it is difficult for the human operator to demonstrate speed and gait choices at the same time. Meanwhile, previous studies in animal [11] and robot [12, 13] locomotion have revealed a close connection between speed and gait choices. Inspired by this discovery, we simplify the demonstration and learning process by designing a heuristic-based *gait selector*, which computes the appropriate gait parameters based on desired forward speed.

**Gait Parameterization**   In our design, each gait is parameterized by three parameters, stepping frequency (SF), swing foot height (SH), and base height (BH). The **stepping frequency (SF)** determines the number of locomotion cycles each second. Similar to [12], we adopt a phase-based parameterization for gait cycles, where each leg alternates between swing and stance. In addition, we assume a trotting pattern for leg coordination, where diagonal legs move together and are $180°$ out-of-phase with the other diagonal. The trotting pattern is known for its stability, thereby being the default gait choice in most quadrupedal robots [28, 14]. The **swing foot height (SH)** determines the leg's maximum ground clearance in each swing phase. While a higher swing height improves stability on uneven terrains by preventing unexpected contacts, a lower swing height is usually necessary for high-speed running. The **base height (BH)** specifies the height of the robot's center-of-mass. While a low base height gives better stability at high speeds, a higher base height can be beneficial when traversing through unknown obstacles.

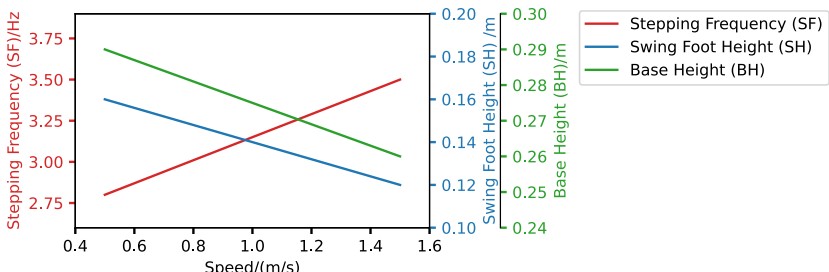

Figure 3: The gait selector selects gait parameters (SF, SH and BH) based on desired forward speed. For example, when the desired speed is 0.5m/s, the speed selector would choose a stepping frequency of 2.8Hz, a swing foot height of 0.16m, and a base height of 0.29m.

**Speed-Based Gait Selection** We use empirical evidence to design the speed-based gait selector, which finds a gait with high traversability for each speed. More specifically, for the boundary speeds (0.5m/s and 1.5m/s), we first try different SFs with a nominal SH (0.12m) and BH (0.26m), and find the lowest SF that would still ensure base stability (2.8Hz and 3.5Hz). After that, we sweep over different values of SH and BH to find the highest value of both that would allow the robot to walk robustly without falling. Lastly, we linearly interpolate the parameter values between the boundary speeds to find the gait for intermediate speeds. See Fig. 3 for details.

# 6 Low-level Convex MPC Controller

The low-level convex MPC controller computes and applies torques to each actuated degree of freedom, given the locomotion skills from the skill policy. Our low-level convex MPC controller is based on Di Carlo et al. [28] with two important modifications. Firstly, due to the robot's small form factor, it needs to constantly re-orient its body on uneven terrains, such as bumps and potholes. Therefore, we implemented a state estimator to estimate the ground orientation and adjust the robot pose to fit the ground, similar to Gehring et al. [31]. Secondly, to reduce foot slipping, we implement an impedance controller for stance legs [32]. In addition to the motor torque command computed by MPC, the impedance controller adds a small feedback torque to track the leg in its desired position. We found both techniques to improve locomotion quality significantly. Please refer to Appendix A for details.

# 7 Experiment and Result

To see whether our framework can learn to adapt locomotion skills based on terrain semantics, we deploy it to a quadruped robot and test it in a number of outdoor environments in the real world. We aim to answer the following questions in our experiments:

1. Can our framework operate without failure in complex offroad terrains for extended periods of time, and how does it compare with existing baselines?

2. How does our framework generalize to terrain instances not seen during training?

3. Can our framework walk at high speed while ensuring safety?

4. What are the important design choices in training the perception module, and how does it affect performance?

## 7.1 Experiment Setup

We implement our framework on an A1 quadrupedal robot from Unitree [14]. We equip the robot with an Intel Realsense D435i camera to capture RGB images and a GPS receiver to track its real-time location. We implement the entire control stack in the Robot Operating System (ROS) framework [33] and deploy it on a Mac Mini with M1 chip, which is mounted on the robot. The convex MPC controller runs at 400Hz, and the speed policy and gait selector run at 3Hz.

To train the speed policy, we collected 7239 frames of data on a variety of terrains, which corresponds to 40 minutes of robot operation. The entire process, including robot setup, data collection,

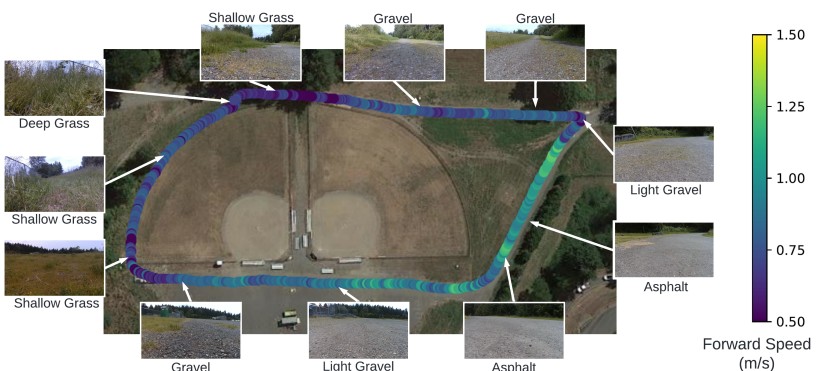

Figure 4: The 450m-long test trail consists of multiple terrain types such as deep grass, shallow grass, gravel, and asphalt. The learned skill policy adjusts the speed and gait based on terrain semantics and walks faster on easier terrains.

| Policy Type | Speed (m/s) | Gait Params (SF, SH, BH) | Traversal Time (min) | Number of Failures |
|---|---|---|---|---|
| Fixed-Slow | 0.5 | [2.8, 0.16, 0.29] | 15 | **0** |
| Fixed-Mean | 0.8 | [3.0, 0.15, 0.28] | ∞ | 3 |
| Fixed-Medium | 1 | [3.1, 0.14, 0.28] | ∞ | 4 |
| Fixed-Fast | 1.5 | [3.5, 0.12, 0.27] | ∞ | 10+ |
| Speed-Only | Adaptive | [3.1, 0.14, 0.28] | ∞ | 9 |
| Gait-Only | 0.8 | Adaptive | ∞ | 2 |
| Unitree-Normal | Tele-operated | N/A | 11±0.4 | **0** |
| Unitree-Sport | Tele-operated | N/A | ∞ | 2 |
| Fully-Adaptive (ours) | Adaptive | Adaptive | **9.6±0.2** | **0** |

Table 1: Performance of different policies on the test trail (450m). Compared to other policies, our framework completes the entire trail without failure in the shortest time. We repeat the Unitree-Normal and Fully-Adaptive policies 3 times and report the mean and standard deviation of the traversal time. We do not repeat the other policies due to excessive robot damage.

and battery swaps, took less than an hour. The speed policy is trained on a standard desktop computer with an Nvidia 2080Ti GPU, which took approximately 20 minutes to complete.

## 7.2 Fast and Failure-free Walking on Multiple Terrains

To evaluate the adaptivity of our framework, we test our framework on an outdoor trail with multiple terrain types, including deep grass, shallow grass, gravel, and asphalt (Fig. 4). Our controller switches between a wide range of skills as it traverses through the trail, from slow and careful stepping to fast and active walking, and completes the 450m-long trail in 9.6 minutes, comparable to the performance of human demonstrations (10 minutes).

We compared our learned framework with the following baselines on the same test trail (Fig. 4), including Unitree's built-in controllers and variants of our controller with no or limited adaptation. The result is summarized in Table. 1. Please refer to Appendix. B.2 for further details.

**Unitree's built-in controllers**    We tested two modes of the built-in controller, a *normal* mode (Unitree-Normal) that walks up to 1m/s, and a *sports* mode (Unitree-Sport) that walks up to 1.5m/s. Both controllers do not include perception and assume a fixed gait at all times. Although normal mode completed the entire trail without failure, it walked slower than our learned framework, especially on asphalts, due to limitations on the maximum speed. On the other hand, the sports mode controller failed to complete the course and got stuck in deep grass twice due to insufficient swing foot clearance.

**Fixed Skill with No Adaptation**    For these baselines, we disabled the perception module and operated the robot with a fixed locomotion skill. We tested four skills, namely slow, mean, medium, and fast, operating at 0.5m/s, 0.8m/s, 1m/s, and 1.5m/s, respectively, with the corresponding gait selected according to Fig. 3. The slow, medium and fast skills cover the range of possible speeds achievable by our low-level controller, and the mean skill walks at speed similar to the average speed

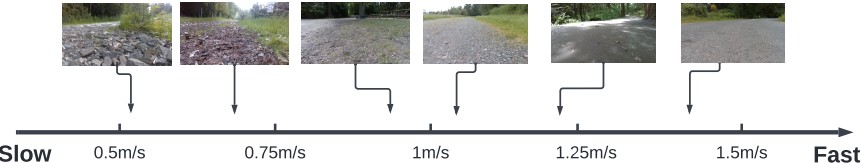

Figure 5: Desired speed computed by the skill policy. The policy prefers faster skills for rigid and flat terrains
and prefers ~~slower~~ ~~skills for deformable or uneven terrains.~~

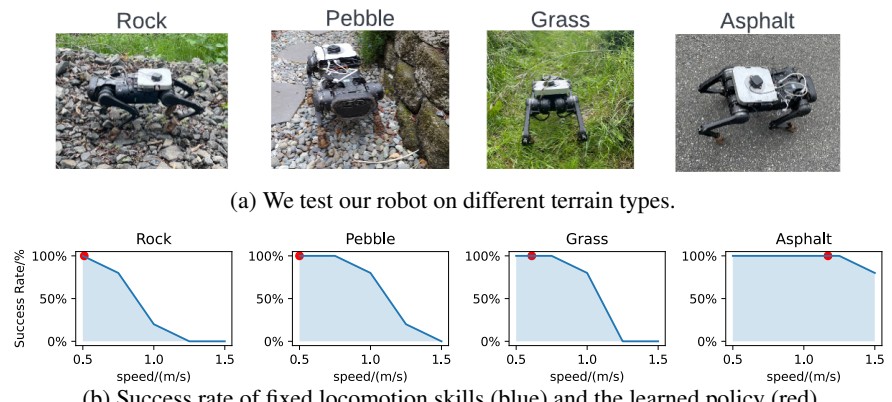

(a) We test our robot on different terrain types.

(b) Success rate of fixed locomotion skills (blue) and the learned policy (red).

Figure 6: Our framework learns fast and safe locomotion skills. *Top:* We deployed our skill policy to 4 different
terrains. *Bottom:* Our policy finds a high speed in the safe region of each terrain.

achieved by our adaptive policy (0.78m/s). The mean, medium, and fast skills failed to complete the
trail and incurred failures. While the slow skill completed the trail without failure, its traversal time
is 50% longer than our learned framework.

**Adapt Speed or Gait Only**    In our framework, we design a robot skill to be a combination of gait
and forward speed. To justify this design, we design two policies, where the robot adapts the gait or
the forward speed only. For the speed-only policy, we fix the gait parameters as if the forward speed
is 1.0m/s in Fig. 3 and adapt the speed using our framework. For the gait-only policy, we fix the
base speed to be 0.8m/s, similar to the average speed attained by our learned policy, and adapt the
gait using our framework. Both policies failed to complete the trail. For the speed-only policy, we
found the fixed gait to only work well when the base speed was close to 1m/s and frequently failed
at either higher or lower speeds. For the gait-only policy, the robot managed to walk through most
of the trail but slipped twice on rocky terrains.

## 7.3    Generalization to Unseen Terrain Instances

To further test the generalizability of our framework, we deploy the robot on a number of outdoor
trails not seen during training. The trails contain diverse terrain types, such as dirt, gravel, mud,
grass, and asphalt. The robot traverses through these test trails without failure and adjusts its loco-
motion skills based on terrain semantics. Please refer to Appendix. B.1 for details. To demonstrate
the skill choices of our framework, we select a few key frames from the camera images and plot
the corresponding speed in Fig. 5. Generally, the skill policy selects a faster skill on rigid and flat
terrains and a slower speed on deformable or uneven terrains. At the time of writing, the robot has
traversed through over 6km of outdoor trails without failure.

## 7.4    Analysis on Speed and Safety

To test the performance of the learned skill policy in terms of speed and stability, we deploy the
learned skill policy on four different terrains, including rock, pebble, grass, and pebble (Fig. 6a). We
compare our semantics-aware skill policy with 5 fixed skills, where the speed linearly interpolates
between 0.5m/s and 1.5m/s. For each speed, the corresponding gait is selected according to Fig. 3.
For each terrain and skill combination, we repeat the experiment 5 times and report the success rate,
where a trial is considered successful if the robot does not fall over during the traversal (Fig. 6b).

By comparing the success rate at different speeds, we obtained an approximation of the safe speed range for each terrain. We then test the performance of our framework by comparing the average speed obtained by our learned skill policy on each terrain against these safe speed ranges.

The maximum safe speed varies significantly on different terrains. For example, while the robot can walk up to 1.25m/s without failure on asphalt, it can only walk up to 0.5m/s on rock, due to unexpected bumps and foot slips on the surface. Although not directly optimized for speed or robot safety, our learned policy finds a close-to-maximum speed in the safe region of each terrain after learning from human demonstrations. We also noted that on pebble and grass, there is a slightly larger gap between the maximum safe speed and the speed selected by the skill policy. One reason for this is that the speed demonstrated by the human operator can be more conservative than the maximum safe speed.

### 7.5 Ablation Study on Perception Module

We compare our way of training the perception-enabled speed policy with a few baselines, which either train the policy from scratch without pre-training or extract the pre-trained embedding directly from the predicted semantic classes. We find that our policy, which is fine-tuned from the output of the hidden layer, achieves the smallest error on the validation set and predicts the speed map with high precision. Please see Appendix B.3 for further details.

## 8    Limitations and Future Work

In this work, we present a hierarchical framework to learn semantic-aware locomotion gaits from human demonstrations. Our framework learns to adapt locomotion skills for a variety of terrains using 40 minutes of human demonstration and enables a robot to traverse over 6km of outdoor terrains without failure. One limitation of our framework is that, while our robot walks robustly on a variety of off-road terrains, its performance is limited by the low dimensionality of human demonstrations. For more difficult terrains such as steps or gaps, the robot will need more agile behaviors such as jumping, which requires a deeper integration between the perception system and low-level motor controller and learning more skills than speed or gait demonstrations. Another limitation is that the perception system assumes that there is no non-traversable obstacles ahead of the robot and therefore does not adjust the heading of the robot. In future work, we plan to increase the agility of our controller and integrate path planning into our framework so that the robot can operate fully autonomously in challenging off-road environments.

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
