# OpenReview forum: "Learning Semantics-Aware Locomotion Skills from Human Demonstration"
_robot-learning.org/CoRL/2022/Conference — CoRL 2022 Poster_

### Official Review · Reviewer_ofNu · 2022-07-20

**Originality:** Good
**Technical Quality:** Very Good
**Clarity Of Presentation:** Excellent
**Impact:** 3

**Recommendation:**

Strong Accept: I recommend accepting the paper and will argue for my recommendation even if other reviewers hold a different opinion.

**Summary:**

The paper presents a quadruped robot locomotion method, where MPC-based locomotion is enhanced with visual-based terrain type identification that drives correctly speed and gait. The whole method learns from human demonstrations on the real hardware.

**Issues:**

All weaknesses above need to be handled. Most are on the writing part and I think one is on the experimental part (slopes).

**Quality Of The Limitations Section:**

Limitations are addressed clearly

**Reviewer Expertise:**

5: The reviewer is absolutely certain that the evaluation is correct and very familiar with the relevant literature

**Robotics Focus:**

Sufficient demonstration on hardware

**Strengths And Weaknesses:**

Strengths:
- This is a clever work. It is a basic concept, i.e., when moving on different terrains, the speed and gait might need to adapt accordingly.
- The technical part is easy to follow and sound -- having said that the development part is not extremely challenging.
- The experiments are overall nice, with real-world locomotion on various terrains.

Weaknesses:
- Abstract: "close-to-optimal speed" is a bit of an exaggeration. It has not been shown anywhere in the results what is the optimal, and thus this sentence is not supported. Moreover, I have the feeling that it is also not true.
- Intro/RW: the intro and later the related work, lacks some work on terrain semantic classification for locomotion, e.g., "Terrain Classification and Locomotion Parameters Adaptation for Humanoid Robots Using Force/Torque Sensing", by K. Walas et al. (IJHR'16) for proprioception and "Terrain Segmentation and Roughness Estimation using RGB Data: Path Planning Application on the CENTAURO Robot", by V. Suryamurthy et al. (IJHR'19) for exteroceptive segmentation.
- Intro: "human expert demonstration" there is a big discussion now how those humans became experts and what does this mean?
- Intro: "Fast and Failure-free: this is again very vague and generic. In best case, terrain types need to be mentioned at least. Otherwise, this does not have any particular meaning.
- RW: In the end of the RW section, the selection to learn from real demonstration is highly proposed. Although, we know that learning in simulation (e.g., in isaac) achieves very good results without the need to adapt anything on the physical robot (e.g., the work on ANYmal in the latest Science publication). It needs to be justified better why in this case simulations could not do the job.
- Sec 4: "instead, we extract a semantic embedding" this is very vague and needs to be explained further. Why the semantic segmentation is not good, but the embedding is? Why it is feeded via a fully connected layer? More justification is needed.
- Sec. 4: "1m long, 0.3m wide in front of the robot." what if the camera changes orientation or position. How much of the method can work as is?  How much retraining is needed?
- Sec. 4: "using imitation learning": it is still confusing how the humans decided the "optimal" speed for each type of terrain.
- Sec. 5: the speed-gait selection seems to be very tedious and not extendable easily to other robots. Any alternative? Any less heuristic approach?
- Sec. 6: slope handling is claimed but none of the experiments include slopes. Not sure if it is good to be mentioned if it was not tried at all.
- Sec. 7: you compare with Unitree's controller, but in reality [21] should be your baseline. How much better than [21] do you perform?
- Sec. 7: " it can only walk up to 0.5m/s on rock", how was this evaluated? What if different gaits were applied?
- Sec. 7.3: "traverses through all of them without failure", this is very generic and probably not true. Unless the authors claim we have found the optimal MPC-based control to deal with all terrains. In other words, a specification of gait and terrain type is needed to justify this.
- General: What happens when you traverse from one terrain type to another? How the robot adapts to the new gait and speed? What happens when terrain segmentation is noisy, and what happens when the robot walks on a misclassified terrain?

**Summary Of Recommendation:**

I believe the work requires some improvement on the weaknesses mentioned above, but in general it is nice. The main problem is overselling the approach. But if we ignore this part, the paper appears to be solid, sound, easy to follow, and nice.

---

> ### Author Response · Authors · 2022-08-19
> **Response to Reviewer ofNu**
>
> Thanks for the detailed comments. We are working on a revision of the paper, where we will address the reviewer’s comments on writing and state the claims of the paper more accurately. In the meantime, please find our answer to the specific questions below:
>
> > Slope handling is claimed but not tested in the experiments.
>
> Sorry for the confusion. By “slope”, we mean small patches of uneven terrain where the robot is required to stand with non-zero roll and pitch angles, such as rocks and potholes. The robot incurred these scenarios frequently during testing, and the slope handling component is important in stabilizing the robot. To avoid confusion, we will rename “slope detection” to “uneven terrain detection” in the revision.
>
> > The infeasibility of simulation in our task needs to be further justified.
>
> We find it difficult to leverage simulation for our task for two reasons. Firstly, while it is generally easier to simulate the dynamics and geometric perception (depth cameras or lidars) of the robot, accurately simulating the RGB rendering and terrain semantics is still currently difficult. Secondly, the complexity in off-road environments poses an additional challenge for the simulator, because many scenarios in off-road locomotion, such as legs getting trapped in deep grass, can be difficult to model accurately. We will clarify this in the revision.
>
> > The use of semantic embedding and fully-connected layers needs to be further justified.'
>
> We chose to use the embedding, instead of the predicted semantic score, to better handle the distributional shift between the perception module’s training and testing data. Since we train the semantic segmentation network on an off-road driving dataset, it experiences distributional shifts and sometimes misclassifies terrains when tested on robot images. However, we find that the embedding still serves as a good feature vector for the terrain semantics. We use fully-connected layers on top of the embedding in order to generate the pixel-wise speed map. While the speed map is only used to compute the average speed command in this paper, we plan to use it for path planning in future works. We will clarify this in the revision.
>
> > Is the perception module sensitive to changes in camera orientation?
>
> Since we compute the speed command by averaging over a fixed region of the speed map, small changes to camera orientations does not matter, as long as the region still captures the terrain right in front of the robot. In our experiments, we use the built-in camera mount from the robot manufacturer, which allows rotation along the pitch axis and sometimes changes the camera’s pitch angles during walking. We do not set the camera to a specific pitch angle between trials, and do not find the rotations in the camera's pitch angle to affect the performance of the perception module.
>
> > The speed-gait selection is platform-specific and can be difficult to extend to other robot platforms.
>
> While our speed-based gait selector is specifically designed based on the A1 robot, the general relationship between speed and gait is observed in both animal and legged robots. We leverage this scientific discovery to simplify the demonstration collection and learning process. For other quadrupedal robots, a similar gait selector can be designed based on its weight and kinematics.
>
> > In addition to Unitree’s built-in controller, controllers in [21] should be compared with as baselines.
>
> We would like to clarify this misunderstanding. We compare our perceptive controller with both the controller proposed in [21] and the Unitree’s built-in controllers. Since we implement the low-level controller based on [21], the fixed-speed, fixed-gait controllers in the ablation study (i.e. fixed-slow, fixed-medium and fixed-fast in Table.1) corresponds to running the convex MPC controller [21] at different speed and gaits. Compared with our perceptive controller, the non-adaptive controllers in [21] either walked too slowly, or incurred failures. We include the Unitree controllers as an additional, third-party baseline to further demonstrate the performance of our controllers.
>
> > What happens when the robot traverses from one terrain type to another, and how does the robot adapt to the new gait and speed?
>
> When the robot traverses to a new terrain type, the low-level MPC controller will receive new speed and gait commands from the skill policy, and computes motor torques based on the new commands immediately. Since we average the speed estimation over a fixed region in the image, the speed and gait change is gradual, and does not lead to sudden perturbations to the low-level controller.

---

> ### Author Response · Authors · 2022-08-20
> **Response, cont'd**
>
> > What happens when terrain segmentation is noisy, and what happens when the robot walks on a misclassified terrain?
>
> As mentioned above, due to the distribution shift between the training and testing data, the segmentation network can sometimes generate wrong or noisy predictions on the robot’s camera image. To avoid the noise and error in predicted class scores, we compute the speed estimation from a semantic embedding, which is less susceptible to distributional shifts.

---

> ### Author Response · Authors · 2022-08-26
> **Revision Posted**
>
> Thanks again for the detailed and thoughtful comments. We have posted our revision to better address your comments. Please refer to the attached PDF for the diff file and the post at the top thread for the compiled revision. The major updates in the revision include:
>
> __1. Added an ablation study on the use of semantic embedding.__
>
> In response to the comment about fine-tuning over the semantic embedding, not the predicted semantic class, we included an ablation study that compares the two approaches. We found that, due to the distribution shift between the training and testing data, the pre-trained segmentation network can generate noisy semantic labels on the robot data, which affects the performance in fine-tuning.  Please see Section 7.5 and Appendix B.3 for more details.
>
>
> __2. Better explained the motivation of real-world training and learning from human demonstrations.__
>
> We have explained in more details about the unavailability of off-road simulation in the related works section. As for the motivation of using human demonstrations, we found that although finding the correct speed for each terrain requires a lot of trial and error, a human operator can easily tele-operate the robot through most terrains at a decent speed. This is likely due to the operator’s previous experience with the robot platform and the low-level controller. We have included this in the motivation for imitation learning in the revised paper.
>
> __3. Included more papers in the related work.__
> Thanks for pointing out the related works on terrain semantic classification for locomotion. We have included them in the related works section in the revised paper.
>
> __4. Rephrased claims about “optimality” and “failure-free”.__
> We have revised several wordings about optimality and failure-free in the revision in response to the comments.
>
> Please let us know if you have any questions.

---

### Official Review · Reviewer_FzJu · 2022-07-31

**Originality:** Excellent
**Technical Quality:** Excellent
**Clarity Of Presentation:** Excellent
**Impact:** 4

**Recommendation:**

Strong Accept: I recommend accepting the paper and will argue for my recommendation even if other reviewers hold a different opinion.

**Summary:**

This paper presents a framework to learn a set of perceptive locomotion skills. The key idea of this work is to leverage existing offroad driving dataset for quadrupedal offroad navigation tasks.

**Issues:**

Aside from the weakness mentioned, several comments on the details of writing:
- Line 25: Here, although it seems intuitive, lacks citations from either perception literature or previous works to support this claim. For example, maybe JJ Gibson's Affordance chapter could support this claim.
- Section 6: Since the MPC controller is from previous works, I suggest putting it into a single paragraph instead of a single section.
- Line 172: "design" or "implement"? Maybe authors say "design" because they are different from how previous works implement impedance control for locomotion. I am confused if this is a novel design or an adaptation from a previous approach.
- Line 182: I think unseen terrains refer to unseen "instances" of terrain types. It's ambiguous about "unseen terrains."
- Table 1: I think it's important to say what's the length of a single trail, as the authors only report a total number of lengths (6km) while the traverse time says around 10 min, which was quite surprising at first. Stating the length of trails themselves would help readers understand the result much clearer.


**Quality Of The Limitations Section:**

Additional details required

**Reviewer Expertise:**

5: The reviewer is absolutely certain that the evaluation is correct and very familiar with the relevant literature

**Robotics Focus:**

Sufficient demonstration on hardware

**Strengths And Weaknesses:**

Strengths:
- A very clean and neat framework to enable perceptive locomotion skills
- The use of existing datasets from other domains for the target downstream tasks
- Both quantitative and qualitative evaluation of the approach, especially the robustness of the models.


Weakness:
- Line 96: Does this mean that authors assume there are no obstacles in terrain that would stop robots from traversing at all? The limitation section mentions future work of navigation, but that would require the robot to reorient its heading directions. This should also be mentioned in the limitations.
- Line 122: Authors should provide more intuition on why using FC on each pixel instead of using convolutional layers over pixels and directly output the forward speed. What is the intuition behind having this intermediate speed map?
- For real robot videos, it would be even better if there is a list of average speed over different domains. When watching the multi-terrain type video, it’s hard to tell if robots are executing at the same speed as individual evaluation.
- Also, since the locomotion skills are not high-speed behaviors, it’s still difficult to tell the speed differences by playing each video individually. Maybe putting all video clips on one single canvas would give a better idea of the gaits’ differences.


**Summary Of Recommendation:**

I strongly recommend to accept this publication with the following two reasons: 1) it shows how to leverage existing datasets for robotics research. It is notoriously known that it’s hard to collect large-scale datasets for robot control as contrast to existing vision or nlp datasets. This work shows that we can leverage large datasets for robot control. And specifically, it’s a novel for the perceptive locomotion community. 2) It shows a working locomotion system that is very impressive. The no-cup video clip of full trail locomotion strongly shows the robustness of the approach. I think it’s quite rare to see in the robotics community nowadays, and it sets up a good example of what is a solid robot learning approach is.

---

> ### Author Response · Authors · 2022-08-19
> **Response to Reviewr FzJu**
>
> Thanks for the thoughtful comments. We are working on a revision of the paper to address the detailed comments from the reviewer. We will also revise the final video to better demonstrate the performance of our controllers.
>
> For the comment about the intuition of using fully-connected layers, we use fully-connected layers instead of the convolutional layers in order to generate the pixel-wise speed map. While the speed map is only used to compute the average speed command in this paper, we plan to use it for path planning in future works. We will clarify this in the revision.

---

> > ### Author Response · Authors · 2022-08-26
> > **Revision Posted**
> >
> > Thanks again for the thoughtful comments. We have posted a revision of the paper to further address your comments. Please see the diff file attached in this post. The fully-compiled revision is also posted at the top thread.
> >
> > In the latest revision, we clearly stated our motivations for using fully-connected layers to generate the speed map, which is to better visualize model predictions and facilitate future studies in navigation and path planning. We have also revised the limitations section to clearly state that we assume there are no non-traversable obstacles in the terrain. Lastly, we have addressed the comments about writing that are listed in the issues section.
> >
> > Please let us know if you have any further questions.

---

### Official Review · Reviewer_Rr19 · 2022-07-31

**Originality:** Fair
**Technical Quality:** Good
**Clarity Of Presentation:** Good
**Impact:** 3

**Recommendation:**

Weak Reject: I recommend rejecting the paper, but will not argue for my recommendation if the majority of other reviewers have a different opinion.

**Summary:**

The paper proposes a system which learns to adapt walking speed based on semantics of terrain. The proposed approach learns, from human demonstration, to regulate forward speed as a function of RGB input. The model is pretrained on a semantic segmentation task and optimized via behavior cloning for the speed control problem. A gait selector is manually designed as a function of desired walking speed. The result is a system which is capable of traversing rough terrain more efficiently than the built-in Unitree controllers.

**Issues:**

Abstract: consider changing "terrain type and property, reveals" to "...properties, reveal"

**Quality Of The Limitations Section:**

Limitations are addressed clearly

**Reviewer Expertise:**

3: The reviewer is fairly confident that the evaluation is correct

**Robotics Focus:**

Sufficient demonstration on hardware

**Strengths And Weaknesses:**

Strengths -

The paper studies an important problem - clearly some amount of semantic information is important for locomotion in offroad environments. Existing works typically fail to adequately address this in that they only model the geometry of the surface rather than material properties like friction, or by being entirely reactive. Thus the paper suggests an interesting innovation.

Weakness -

The learning problem proposed is perhaps too simple - controlling speed alone is likely insufficient for more challenging terrain, as illustrated in recent papers which additionally learn to regulate gait (e.g. the Hutter Science paper cited). It seems as though some combination of these ideas is required for robust locomotion, controlling both the desired forward speed and the parameters of the gait together as a function of anticipated terrain properties and proprioceptive data. That the method never fails on the test domains suggests that perhaps the tests are not sufficiently challenging. Moreover, comparison to a method which is purely geometry-based would be instructive in evaluating whether the semantic module indeed constitutes a substantial improvement in performance in the domain studied (e.g. even a simple variant without the semantic segmentation pretraining).

Additionally, the reliance on human demonstrations may present a challenge in scaling the method.

**Summary Of Recommendation:**

I'm recommending weak rejection on the basis of the simplicity of the learning problem posed and the relatively limited experimental evaluation.

---

> ### Author Response · Authors · 2022-08-19
> **Response to Reviewr Rr19**
>
> Thanks for the detailed comments. Please refer below for our responses. We will incorporate your suggestions and our answers into the revision.
>
> > Learning problem is too simple and controls speed alone
>
> We would like to clarify this misunderstanding. In addition to speed, our method adjusts the gait of the robot (e.g. foot clearance, stepping frequency, etc.) simultaneously. As shown in the ablation study (Table 1), adapting the speed and gait simultaneously is essential for our robot to traverse through off-road terrains. In contrast, adapting speed only or gait only would lead to more failures on the same test trail. Our method selects the speed first based on a learned policy, and selects the corresponding gait based on speed. This is inspired by previous studies in both animal [1] and robot [2, 3] locomotion, where researchers observe a close connection between the gait patterns and forward speed. In this paper, we leverage this connection to simplify data collection for demonstration: The human teleoperator only needs to adjust one scalar (speed), instead of multiple values (speed and other gait parameters) in real time when controlling the robot in challenging environments. Note that the alternative of learning without demonstration for off-road semantic-aware locomotion is difficult due to the lack of accurate simulation and the safety concerns in the real world, as stated in the introduction. While our method may limit the overall expressiveness of the high-level policy, it significantly simplifies the demonstration and learning process without affecting the performance of our robot.
>
> > Comparison with geometry-based approach would be instructive.
>
> We would like to clarify that geometry and semantics provide complementary sets of information that can both be essential for robust off-road locomotion. While a geometric understanding of terrain shapes is important for certain environments such as stairs and stepping stones, a semantic understanding can be more useful for many scenarios tested in this paper, such as deep grass (which is easily misclassified as rigid obstacles from geometric sensors) and pebble (where the slip between pebbles can be overlooked by geometric sensors). A promising direction would be to combine both semantics and geometry, which we plan to explore in future works.
>
> As for the proposed baseline where the policy is trained without semantic segmentation pre-training, we are testing it on our robot, and will report the result in the next iteration of the paper.
>
> > Reliance on human demonstrations may present a challenge in scaling the method
>
> While human demonstration is an essential component of our learning setup, we do not find it as the bottleneck in scaling our method. Our framework is data-efficient: with the pre-trained semantic embedding, our framework can learn to walk in a variety of terrains (Section 7.3) using only 40 minutes of human demonstration data. Moreover, the demonstration data can be easily collected on any off-road terrains, and does not require extra setups such as motion capture. Additionally, learning from human demonstrations enables us to train directly in the real world without incurring significant robot damage. Therefore, we believe that our learning-from-demonstration setup can be scaled to more terrains without significant extra effort.
>
>
> ### References
> [1] Gait and the energetics of locomotion in horses. Hoyt. et al.
>
> [2] Minimizing Energy Consumption Leads to the Emergence of Gaits in Legged Robots. Fu. et al.
>
> [3] Fast and Efficient Locomotion via Learned Gait Transitions. Yang. et al.

---

> > ### Author Response · Authors · 2022-08-26
> > **Revision posted**
> >
> > Thanks again for the detailed feedback. We have posted a revision to further address your comments. Please see the attached pdf for the diff file and the latest thread for the compiled revision. The major changes include:
> >
> > __1. An ablation study on the training of the perception module.__
> >
> > We conducted an ablation on the training of the perception module. More specifically, we compare our method of fine-tuning from a pre-trained semantic embedding with a policy trained from scratch without pre-training. We find that the policy trained from scratch incurred a larger validation error and cannot predict the speed map with fine granularity. Please see section 7.5 and appendix B.3 for further details.
> >
> >  __2. Updated limitations section with a discussion about challenging terrains.__
> >
> > We have updated the limitations section to mention that our framework cannot handle terrains with large discontinuities (high steps, gaps, etc.), which requires more agile behaviors such as jumping. We will look into this in future studies.
> >
> > Please let us know if you have any questions about the latest revision.

---

### Official Review · Reviewer_oYTQ · 2022-08-01

**Originality:** Very Good
**Technical Quality:** Good
**Clarity Of Presentation:** Good
**Impact:** 3

**Recommendation:**

Weak Accept: I recommend accepting the paper, but will not argue for my recommendation if the majority of other reviewers have a different opinion.

**Summary:**

This paper presents an architecture for adapting the locomotion of a legged robot in response to semantic terrain information. A human teleoperates the robot with a joystick to provide scalar velocity labels associated with different terrains. A mapping from RGB images to velocities is learned using the features of a pretrained vision module to promote generalization. Then, velocity commands are heuristically mapped to gait parameters, and the resulting locomotion skill is actuated using an MPC controller from prior work. The complete system enables the robot to automatically run faster on easier terrains and walk carefully on harder terrains. The system can complete novel trails in this manner without failure.

**Issues:**

- A main result of the paper, the failure rate in Table 1, is only reported for one trial per policy. This metric seems to have high variance and is also exposed to influence by the human operator. For example, the "unexpected bumps" you refer to in Section 7.5, which I read as small but dangerous features like potholes, would cause high variance in the failure rate depending on whether the robot happens to step on them or not in that trial. Do you have some video showing those failures? Can you report more trials or propose a less brittle evaluation metric?
- Also in Table 1, the Fixed-Medium controller has a much higher average speed than the Fully-Adaptive controller. It seems fair to add an evaluation of a fixed controller with equal avg speed (looks like 0.78m/s?)
- Some potential limitations of this framework should be discussed in the paper. The low-level policy is limited in how it can adapt to terrain properties. For example, there is a fixed friction value in the model used for MPC controller. Consider that if this were tuned for each terrain, the locomotion on slippery terrains could be stable at higher speeds. But it's impractical for the demonstrator to provide a demo that captures this, or in general to demonstrate adaptation in anything more than one or two highly interpretable dimensions of gait.
- During a discrete transition between terrains, something interesting happens. If transiting from challenging to easy terrain, the robot needs to speed up *after* it leaves the challenging terrain. If transiting the opposite way, from easy to challenging terrain, the robot must slow down *before* leaving the easy terrain. It seems like this behavior cannot be captured by your approach which is based on averaging in a fixed region. Please elaborate on this.

**Quality Of The Limitations Section:**

Additional details required

**Reviewer Expertise:**

5: The reviewer is absolutely certain that the evaluation is correct and very familiar with the relevant literature

**Robotics Focus:**

Sufficient demonstration on hardware

**Strengths And Weaknesses:**

Strengths: The perception of semantic terrain information is a promising research area for locomotion, and this paper is the earliest known functional proposal. The proposed architecture is novel and demonstrates nice integration of several control components.

Weaknesses: A main conclusion of the paper is drawn from a single trial of a high-variance metric.

**Summary Of Recommendation:**

The perception of semantic terrain information is a promising research area for locomotion, and this paper is the earliest known functional proposal. The proposed architecture is novel and demonstrates nice integration of several control components. However, a main conclusion of the paper is drawn from a single trial of a high-variance metric. The authors should either (a) increase the number of trials and provide details about each failure, or (b) choose a less brittle evaluation metric. Also, the architectural choices in the paper may have some limitations, which the authors should express clearly to invite follow-up work in this area.

---

> ### Author Response · Authors · 2022-08-19
> **Response to Reviewer oYTQ**
>
> Thanks for the detailed feedback. We are working on a revision of the paper to address the comments. In the meantime, please see below for our responses to the specific questions.
>
> > The main conclusion of the paper is drawn from a single trial of a high-variance metric.
>
> For the results in Table 1, we have tested our fully-trained controller 4 times across the test trail. For all the trials, the robot completes without failure using a similar amount of time (9.5-10 minutes). We did not repeat our baselines multiple times on the same trail, due to excessive hardware damages from failures and heavy battery drains from slow policies.
>
> > It is fair to add an evaluation of a fixed controller with equal avg speed (0.78m/s).
>
> Thanks for the suggestion. We will include this in the revision of the paper before the discussion phase ends.
>
> > Our framework can only learn one or two highly interpretable dimensions of gait
>
> We agree that the learning-from-demonstration setup can limit the overall expressiveness of the framework. We will include a discussion about this in the limitations section in the revision.
>
> > How does the proposed framework handle discrete transitions between terrains?
>
> In our setup, the robot selects the locomotion skill based on future terrain right in front of the robot, which may be different from the terrain that the robot is currently stepping on. As is pointed out, this mismatch between current and future terrain can lead to a short period of incorrect skill selection at discrete terrain transitions. However, in practice, this mismatch does not significantly affect the robot’s stability because the stable low-level MPC controller will handle this transition stage.

---

> > ### Comment · Reviewer_oYTQ · 2022-08-26
> > **Reply from Reviewer oYTQ**
> >
> > Thanks! I am looking forward to the revised manuscript including
> > - baseline fixed controller with equal avg speed
> > - additional discussion of limitations
> >
> > Do the authors plan to upload this soon?
> >
> > If at all possible, I would encourage the authors to conduct one or two more trials of the strongest baseline, and to include video of the times when the robot failed. This would address my concern about variance and I think it would help readers be very confident in the paper's conclusions.

---

> > > ### Author Response · Authors · 2022-08-26
> > > **Revision Posted**
> > >
> > > Thanks again for the detailed feedback and for the wait. We have posted a revision of our paper to further address the comments. Please see the attached pdf for diff file and the pdf at the top of the thread for the compiled revision. Specifically, we have made the following changes in response to your comments.
> > >
> > > __1. Added experiments and analysis on statistical significance.__
> > >
> > > We re-run our policy and the best-performing baseline (Unitree-Normal) baseline 3 times on the same test track and report the standard deviation in Table 1. We also tested a fixed policy at 0.8m/s (close to the average speed achieved by our adaptive policy) on the same trail and included the result in Table 1.
> > >
> > > __2. Added more details about failure cases.__
> > >
> > > For a more detailed analysis of failure cases, we have included the GPS tracking and failure locations of all policies with failures in Appendix B.2. With regard to variance, note that although the heading of the robot is tele-operated, the GPS trackings in Fig.10 do not deviate significantly from each other and cover similar terrain types.
> > >
> > > __3. Revised the limitations section.__
> > >
> > > We have revised the limitations section and explicitly state the limitation in the low-dimensionality of human demonstrations.
> > >
> > > Please let us know if you have any questions about the latest revision.

---

### Author Response · Authors · 2022-08-26
**Diff File for the Revised Paper**

Please refer to the attached PDF for a Latex diff file that highlights the changes we made in the revision.

---

### Author Response · Authors · 2022-08-26
**Revisions posted, Summary of Changes**

Thanks all for the comment. We have posted a revision of the paper to address the reviewer’s concerns. We have also included a diff file that highlights the changes we made in the revision. In summary, the major changes in the revision include:

__1. Added an ablation study on design choices for the perception module.__

In response to the comments from reviewer Rr19 and ofNu, we conducted an ablation study on the design of the perception module and compared our way of training the speed policy with two baselines, including one that trains from scratch without pre-training and one that uses the predicted segmentation class as the embedding. We found that both baselines cannot perform as well as our method. Please see Section 7.5 and Appendix B.3 for details.

__2. Repeated experiments for statistical significance and added analysis on failure cases.__

In response to the comments from reviewer oYTQ, we re-evaluated the policies multiple times on the same test trail and updated the results with statistical significance in Table 1. We also tested a fixed skill at the average speed achieved by the perceptive policy on the same test trail. Lastly, we added the GPS trackings and failure locations of all failed policies in Appendix B.2.

__3. Included more details in the limitations section.__
In response to the comments from reviewer oYTQ and FzJu, we have updated the limitations section to explicitly mention that our framework relies on low-dimensional human demonstrations and assumes there are no non-traversable obstacles.

__4. Re-organized several sections for clarity.__

For clarity, we have re-organized the results section (section 7) for a more consistent flow of results. We have also re-organized related works (section 2) and the section about speed policy (section 4) for clarity.

Please let us know if you have any questions about the latest revision.

---

### Meta-Review · Area_Chair_TWix · 2022-08-15

**Recommendation:** Accept (Poster)
**Confidence:** 5

**Metareview:**

I believe many reviewers are convinced by the author's response. The motions and behaviors of the robot in the video are not impressive but I respect the reviewers' opinion.

Quality: The reviewers agree that the paper's quality is high.

Clarity: All reviewers rated the clarity as good or above. However, a few comments about it in the reviews should be addressed.

Originality: The presented method is quite simple but the application to legged robotics seems original.

Significance: The reviewers provide a detailed list of issues. There are a few critical ones that the authors should answer in detail, especially the ones by the reviewer Rr19. First, changing the forward speed alone is insufficient to deal with many wild environments. The paper narrowed the scope of learning too much and the performance of the controller is highly limited. The accompanying video is not as impressive as those of the recent related works. The robot probably could run on most of the terrains if it used a better lower-level controller. Second, there are many heuristically designed components because the proposed controller is used as a combination of an MPC controller and a learned planner.

**Best Paper Nomination:**

No

---

> ### Author Response · Authors · 2022-08-19
> **Response to Meta Review**
>
> Thanks for the comment. We would like to emphasize that in addition to speed, our method adjusts the gait of the robot simultaneously, which is important for the robot’s offroad traversability (See ablation study and Table 1 for further details).
>
> As for the heuristically designed components, our method selects the speed first based on a learned policy, and selects the corresponding gait based on speed. The speed-based gait selector is inspired by the speed-gait relationship, which is observed in both animal [1] and robot locomotion [2, 3]. Leveraging this relationship simplifies the demonstration collection and training process. The low-level MPC controller is effective and widely used in legged locomotion [4-6]. Building on top of this powerful MPC framework, we learned an additional high-level vision policy to significantly improve the robot’s traversability in a wide variety of terrains.
>
>
> [1] Gait and the energetics of locomotion in horses. Hoyt. et al.
>
> [2] Minimizing Energy Consumption Leads to the Emergence of Gaits in Legged Robots. Fu. et al.
>
> [3] Fast and Efficient Locomotion via Learned Gait Transitions. Yang. et al.
>
> [4] Dynamic Locomotion in the MIT Cheetah 3 Through Convex Model-Predictive Control. Di Carlo. Et al.
>
> [5] Highly Dynamic Quadruped Locomotion via Whole-Body Impulse Control and Model Predictive Control. Kim. et al
>
> [6] Feedback MPC for Torque-Controlled Legged Robots. Grandia. et al